# Genomic characterization of *Salmonella enterica* isolates causing typhoid among Ghanaian patients

Emmanuel K. Sam[1,2], Derrick A. Daah[1,2], Doreen Akorwome[2], Righteous K. Agoha [2], Enock K. Amoako[1], Collins M. Morang'a[1], Lucas N. Amenga-Etego[1]*, Samuel Duodu [1,2]*

**1** West African Centre for Cell Biology of Infectious Pathogens, College of Basic and Applied Sciences, University of Ghana, Legon-Accra, Ghana, **2** Department of Biochemistry Cell and Molecular Biology, University of Ghana, Legon-Accra, Ghana

\* saduodu@ug.edu.gh (SD); lamengaetego@ug.edu.gh (LNAE)

## Abstract

*Salmonella enterica* serovar Typhi (*S.* Typhi) is a leading cause of typhoid fever, significantly impacting morbidity and mortality in Ghana. However, genome-resolved data on circulating typhoidal strains remains scarce. We conducted this study to explore the genetic diversity, virulence, and antimicrobial resistance (AMR) profiles of *S. enterica* strains isolated from clinically diagnosed typhoid patients to inform targeted management and surveillance strategies. Twenty-eight *S. enterica* isolates recovered from stool and blood cultures were confirmed by PCR targeting 211 bp amplicon of the *bcfD* gene. Whole-genome sequencing was performed on all isolates followed by multi-locus sequence typing (MLST), SPIFinder, ResFinder, and phylogenetic analyses to characterize sequence types (STs), virulence markers, including antibiotic resistance genes and to define clonal relatedness. The GenoTyphi program was used to assign isolates within known *S.* Typhi lineages. For regional comparison, publicly available *S. enterica* genomes and their corresponding metadata were retrieved from the Bacterial and Viral Bioinformatics Resource Center (BV-BRC) and Pathogenwatch and included in the phylogenetic analysis. MLST revealed significant genetic diversity, with *S.* Typhi ST02 and *S.* Typhimurium ST19 and ST313 being notable. ST02 isolates, predominantly linked to typhoidal strains, formed distinct clusters with isolates from neighboring West African countries, indicating regional transmission dynamics. ST313, associated with invasive infections, was isolated from stool samples. The study identified a high prevalence of virulence genes such as *invA*, *invE*, *sopB*, *sopD*, *cdtB*, *pltA*, and *pltB* among STs that are not implicated in typhoidal salmonellosis. Plasmid analysis showed limited diversity, with plasmid replicons detected in only a subset of isolates (n = 12/28, 42.9%). Plasmid carriage was universal in *S.* Typhimurium (4/4, 100%), dominated by IncFIB(S) and IncFII(S), with IncQ1 also detected in two isolates. In contrast, plasmids were rare in *S.* Typhi (1/6, 16.7%), where only a single isolate harbored IncFIB(S) and IncFII(S).

**Data availability statement:** Genome sequence data generated during this study are deposited in the NCBI database and can be accessed at https://www.ncbi.nlm.nih.gov/bioproject/PRJNA1309152.

**Funding:** This research was supported by WACCBIP, University of Ghana, which receives funding from World Bank African Centre of Excellence Masters/PhD fellowship (ACE02-WACCBIP + NCDs, Awandare) and NIHR funded Global Health Research Group on Establishing Regional Hubs for Genomic Surveillance in West Africa grant reference number NIHR13471. The funders had no role in study design, data collection and analysis, decision to publish, or preparation of the manuscript.

**Competing interests:** The author(s) declare that there are no conflicts of interest.

Other non-typhoidal serovars (n = 18) showed moderate carriage (7/18, 38.9%) and greater plasmid diversity. In addition to IncF-type replicons, these isolates harbored Col(pHAD28), IncN, and IncFIB(K). Antibiotic resistance genes were detected at low frequencies, including *bla*TEM variants (beta-lactam resistance, n = 3), *qnr* (quinolone resistance, n = 3), *sul1/sul2* (sulfonamide resistance, n = 3), *aadA1*, *aph(6)-Id*, *aac(3)-IIa* (aminoglycoside-modifying enzymes, n = 3), *tet(A)* (tetracycline resistance, n = 2), *dfrA1* (trimethoprim resistance, n = 1), and *catA1* (chloramphenicol resistance, n = 1), highlighting limited antimicrobial resistance potential in the populations sampled. The serovars identified were *S*. Typhi (n = 6), *S*. Infantis (n = 4), *S*. Virchow (n = 4), *S*. Chester (n = 2), *S*. Jukestown (n = 1), *S*. Durham (n = 4), *S*. Typhimurium (n = 4), *S*. Wein (n = 1), *S*. Bangui (n = 1) and *S*. Saintpaul (n = 1). Phylogenetic analysis positioned the isolates recovered from the current study within clades of other regional isolates with global clinical relevance. The *S*. Typhi isolates belonged to lineage 3.2.1, not previously reported in Ghana, and not typically associated with multidrug resistance. This study provides an important insight into the genetic characteristics of *S. enterica* strains associated with typhoid fever among Ghanaian patients. Several non-typhoidal Salmonella (NTS) strains were found to harbor virulence determinants, including toxin-associated genes implicated in typhoid pathogenesis, highlighting their potential clinical relevance.

## Introduction

*Salmonella enterica* is a Gram-negative facultative intracellular bacteria of global health importance causing around 1.3 billion cases of disease annually particularly in low- and middle-income countries [1–3]. Typhoidal and non-typhoidal *Salmonella* are the broad classifications of the over 2600 serovars identified that are clinically relevant in humans [4]. *S. enterica* serovar Typhi and Paratyphi are the major typhoidal serovars causing enteric fever (typhoid). Both serovars are human restricted and often associated with invasive disease [5]. While the non-typhoidal salmonella (NTS) serovars have a broad host-range and generally cause self-limiting gastroenteritis, a subset of these NTS serovars (~5%), especially those found in sub-Saharan Africa have evolved invasive characteristics [6]. Typically, invasive NTS (iNTS) infection is not associated with diarrhea, but presents with symptoms of sustained fever (39–40°C), respiratory defects and hepatosplenomegaly similar to other febrile illnesses such as malaria and typhoid fever making definite diagnosis more complicated [7].

The virulence architecture of *S. enterica* is complex governed by a diverse array of genetic factors and regulatory networks [8]. Central to this virulence determinant is the presence of Salmonella Pathogenicity Islands (SPIs), that encode Type three secretion system (T3SS) effectors such as adhesins, toxins, and surface polysaccharides contributing to its pathogenesis. Also, it may carry several plasmids and core genes that increase resistance to multiple antibiotics [9]. In particular, *S*. Typhi and *S*. Paratyphi possess unique virulent factors, enabling them to cause systemic infections

in humans. Notably, the Vi capsular polysaccharide and the typhoid toxin are critical in their ability to evade the immune system and establish infection [10]. Emerging evidence indicates that certain NTS serovars are acquiring similar virulence determinants, enhancing their capacity to cause invasive diseases resembling typhoidal infections. Specific lineages of NTS, particularly *S*. Typhimurium sequence type 313 (ST313) and *S*. Enteritidis, have been shown to exhibit genetic adaptations that facilitate systemic infection, including the acquisition of genes analogous to those found in typhoidal serovars [11,12]. The convergence of virulence traits between typhoidal and non-typhoidal *Salmonella* underscores bacterial evolution and highlights the potential for NTS strains to acquire enhanced pathogenic capabilities [4]. This phenomenon poses significant public health challenges, particularly in regions like sub-Saharan Africa, where iNTS bloodstream infections contribute substantially to morbidity and mortality [13].

Ghana is a typhoid endemic country with several challenges in the comprehensive typhoid surveillance and disease management contributing to significant underreporting and a poor understanding of the true burden of disease [14]. Clinical diagnosis of typhoid fever relies on non-specific symptoms and serological tests such as Typhidot and the Widal test. In our previous study, we highlighted the sub-optimal performance of Typhidot with high false positive rates recorded for culture negative samples [15]. Since this diagnostic tool detects host antibodies (IgG and IgM) against *S*. Typhi, low specificity could imply cross reactivity by other invasive *S. enterica* serovars. Genomic characterization is therefore essential to improve interpretation of the ambiguities of current *S. enterica* diagnostic tools. This study set out to characterize the genetic diversity, virulence factors, and antimicrobial resistance profiles of *S. enterica* strains isolated from clinically confirmed typhoid cases using a Whole Genome Sequencing (WGS) approach. By placing local isolates in regional and global genomic context, we sought to highlight sequence similarities and uniqueness that will contextualize local epidemiology and support surveillance and control strategies.

## Materials and methods

### Molecular confirmation of *Salmonella enterica* isolates by PCR

*S. enterica* isolates were obtained from patients presenting with typhoid fever in selected health facilities in Ghana [15]. These patients were recruited between 01/04/2023 and 31/08/2023. For each of the 28 *S. enterica* isolates confirmed by biochemical test and API 20-E [15], DNA extracts were prepared using the boiling lysis method [16]. Briefly, 3−4 bacterial colonies from overnight Luria-Bertani (LB) agar cultures were picked using a sterile loop and resuspended in 250 µL of TE buffer (pH 7.5) in sterile Eppendorf tubes The mixture was briefly vortexed and incubated on a heating block pre-heated to 102°C for 10 minutes. Following incubation, the tubes were centrifuged at 4°C for 10 minutes at 15,000 × g. A portion of the supernatant (150 µL) was carefully transferred into a DNase/RNase-free Eppendorf tube and stored at −20°C. Molecular confirmation of *S. enterica* was performed using PCR targeting the *bcfD* gene, a fimbrial operon component which is conserved across *S. enterica*. The amplification employed the primer pair, F-5'-CCGGACAAACGATTCTGGTA-3' and R-5'-CCGACATCGGCATTATCCG-3' designed to amplify a 211-bp fragment of the target gene. PCR reactions were carried out in a 25 µL reaction volume comprising 12.5 µL of GoTaq 2 × Master Mix, 6.0 µL of nuclease-free water, 0.75 µL of each primer (0.5 µM), and 5 µL of DNA template under conditions described by [17] with slight modifications.

### Whole genome sequencing

*S. enterica* genomic DNA was extracted from overnight LB broth cultures using the DNeasy Ultraclean 96 Microbial Kit (Qiagen, Hilden, Germany) according to the manufacturer's instructions. The concentration of DNA extracts was determined using the Qubit 4.0 fluorometer (Life Technologies). An amount of 100 ng total DNA per sample was used for library preparation. DNA sequencing libraries were prepared using the Illumina DNA Prep for tagmentation, PCR and clean-up as per the manufacturers' instruction. Attachment of unique dual indexes and Illumina sequencing adapters to each individual library was performed using the Nextera XT Index Kit v2 kit. Each library was then purified using Agencourt AMPure

XP beads (Beckman Coulter) and the size distribution and library quality was assessed with the Agilent 4200 TapeStation. The barcoded libraries were normalized and pooled at equimolar concentrations based on the TapeStation and Qubit results. The pooled library was diluted to 12.5 pM and spiked with 5% Phix (v3) for sequencing. Sequencing was done on the Illumina MiSeq sequencing machine using 2 × 250 bp paired-end sequencing with the MiSeq® Reagent Kit v3 (600 cycle).

## Analysis of whole genome sequence data

FastQC (v0.11.9) tool was used for sequence data quality assessment whiles trimmomatic (v0.39) was employed for adapter sequence removal, trimming of poor-quality sequences with a Phred score <30. The filtered reads were merged into a single file using PEAR (v0.9.11) [18]. The merged reads were de novo assembled using 48 Unicycler (v0.4.8) and aligned using BLAST [19]. QUAST (v5.3.0) was used to assess the quality of the genome assemblies and multiple alignment files (S1 Table in S1 File) [19]. Mash toolkit (v2.3) was employed to compare assemblies to genomes in a reference library [20]. CheckM (v1.2.4) was used to estimate completeness and contamination of the genomes. For assemblies that failed to meet the high-quality standard (> 90% completeness and < 5% contamination), a length-based filtering protocol using Biopython (v1.81) was implemented and only contigs ≥1,500 bp were retained for subsequent comparative genomic analyses (S2 Table in S1 File). The species were assigned using the Speciator tool (https://cgps.gitbook. io/pathogenwatch/technical-descriptions/species-assignment/). Using allelic and profile definitions from the open access databases of Institute Pasteur, powered by the BIGSdb software [21], multi-locus sequence typing (MLST) was carried out, and resistant genes were identified using the BIGSdb-Pasteur platform. Plasmid replicons were located using PlasmidFinder's Enterobacteriaceae database and Inctyper [22]. Serovar prediction was performed in-silico using EnteroBase MLST-based serotyping, which assigned Salmonella enterica serotypes based on genetic markers. Predicted serovars were used to group isolates for downstream comparative analyses. We used SPI Finder (v2.0) to identify Salmonella Pathogenicity Islands, at default parameters [23]. Virulence genes were identified using ABRicate against the curated virulence factor database [24]. Only hits with ≥95% nucleotide identity and ≥90% gene coverage were considered present to minimize false positives. The presence of the cytolethal distending toxin (cdtB) and pertussis-like toxin subunits (pltA, pltB) in the NTS serovars were verified using a local reference database. To construct the database, we extracted the coding sequences for pltA, pltB, and cdtB genes from the S. Typhi CT18 reference genome (GenBank: NC_003198) using a custom Biopython-based script. The presence of these targeted genes in the study isolates was further validated using BLASTn (2.17.0). The extracted reference sequences were queried against the assembled contigs of each isolate. Hits were filtered based on a minimum identity threshold of 95% and a minimum query coverage of 95%. Gene intactness was assessed by calculating the ratio of the aligned length to the expected reference gene length; hits with a ratio 0.99 were classified as full-length. To analyze the genomic location and surrounding genes of the pltA, pltB, cdtB genes, a custom Python script was used to parse the Prokka-annotated GenBank files and for each validated BLAST hit, a genomic region extending 5,000 bp upstream and downstream of the gene coordinates was extracted. These extracted regions were then visualized using Clinker (v0.0.21) to generate comparative gene cluster alignments.

## Evolutionary relationships inferred from pangenomes

A total of 134 genome sequences of S. enterica from human infections were obtained from different global populations and used for evolutionary relationship analysis. Whole genome sequence data for 106 isolates were downloaded from PATRIC-BVBRC [25] along with meta-data which included Bio sample, accession numbers, sequence type, geographical location etc. Genomes were selected to ensure broad geographic representation, and duplicate or closely related replicate genomes were excluded prior to analysis. For pangenome analysis, the 106 complete genomes and 28 field isolates from the current study were annotated using Prokka v1.14.6 at default parameters [25] to generate General Feature Format (GFF) files. The annotated GFF files were then processed using Panaroo v1.2.10 [26] in strict mode to identify core

genes. Core genome alignments generated by Panaroo were subsequently used for phylogenetic analysis. The alignment was processed using IQ-TREE (v2.2.2.7) to construct a maximum-likelihood tree with 1,000 ultrafast bootstrap replicates [27]. The phylogenetic tree was visualized and edited using iTOLv5 [28], integrating metadata such as sequence types and geographic origins.

### Lineage classification of *S.* Typhi isolates

Lineages for the six Ghana UG *S.* Typhi isolates were assigned with the GenoTyphi SNP genotyping pipeline with default settings [29,30]. The program interrogates lineage-defining SNPs across the genome and assigns isolates to the corresponding GenoTyphi clades or subclades, following the hierarchical nomenclature system described by Wong and co-investigators [30]. Public *S.* Typhi genomes and their accompanying GenoTyphi lineage annotations (n = 48) were downloaded from PathogenWatch [31] with their accession IDs. These sequences were processed through the same pipeline described for the core-genome phylogeny. All analyses were run in a controlled environment using Conda (Python 2.7 dependency), with input files in FASTA format. Outputs included isolate-level genotyping reports with clade assignments, which were subsequently used to compare the distribution of *S.* Typhi lineages in our dataset with previously reported data from Africa. The resulting tree was visualized in iTOLv5 [32], and displayed as an unrooted cladogram.

### Data processing, analysis and presentation

Heatmaps illustrating the presence and absence of virulence and antibiotic resistance genes were generated using the open-source R statistical software [33]. The genetic data (csv format) was loaded into R. For each sample, the virulence and antibiotic resistance gene data was recorded as 0 for absent and 1 for present to create two binary matrices across all samples. The heatmap R package was utilized to visualize the virulence and antibiotic patterns.

### Ethics statement

This study was approved by the Ethics Committee of the College of Basic and Applied Sciences (ECBAS), University of Ghana with approval number ECBAS094/21–22 and the Ghana Health Service (GHS) Ethics Research Committee with approval number GHS-ERC:045/01/23. A written informed consent was obtained from all adult participants and/or parental (or legal guardian) consent for children enrolled into the study.

## Results

### Genomic diversity and phylogenetic relatedness of *S. enterica* isolates

Genomic analysis of the isolates revealed considerable diversity across sequence types, underscoring the heterogeneity of strains in circulation in the study areas. From the MLST data *S.* Typhi ST02 accounted for 21.4% (n = 6) of the isolates and formed a distinct clade with other Ghanaian isolates (grey) and isolates from Kenya, Uganda, and Nigeria. *S.* Typhimurium ST19 (green) comprising 13.8% (n = 4) of the isolates were phylogenetically placed within a well-defined clade containing other *S.* Typhimurium strains from diverse geographical origins (Fig 1). The isolate belonging to ST313 (dark yellow) clustered exclusively with other Ghanaian isolates and isolates from Malawi. This sequence type, known for its association with invasive *Salmonella* infections in sub-Saharan Africa, highlights the regional adaptation of ST313 and its epidemiological significance in Ghana and the sub-region. Furthermore, ST181 (brick orange) formed clades with isolates from neighboring Burkina Faso and Niger, suggestive of Sahelian adaptation. Other sequence types such as *S.* Durham ST3238, *S.* Chester ST1954 and *S.* Infantis ST603 clustered with ST2640 from Angola and unidentified STs from Ghana and Malawi reinforcing the regional dominance of these lineages. For lineage classification, all *S.* Typhi UG isolates (n = 6) were assigned to lineage 3.2.1 using the Genotyphi SNP-typing pipeline. This was quite distinct from other West African *S.* Typhi lineages belonging to 3.1.1, 2.3.2, 2.2, and 4.3., including those previously reported in Ghana (Fig 2).

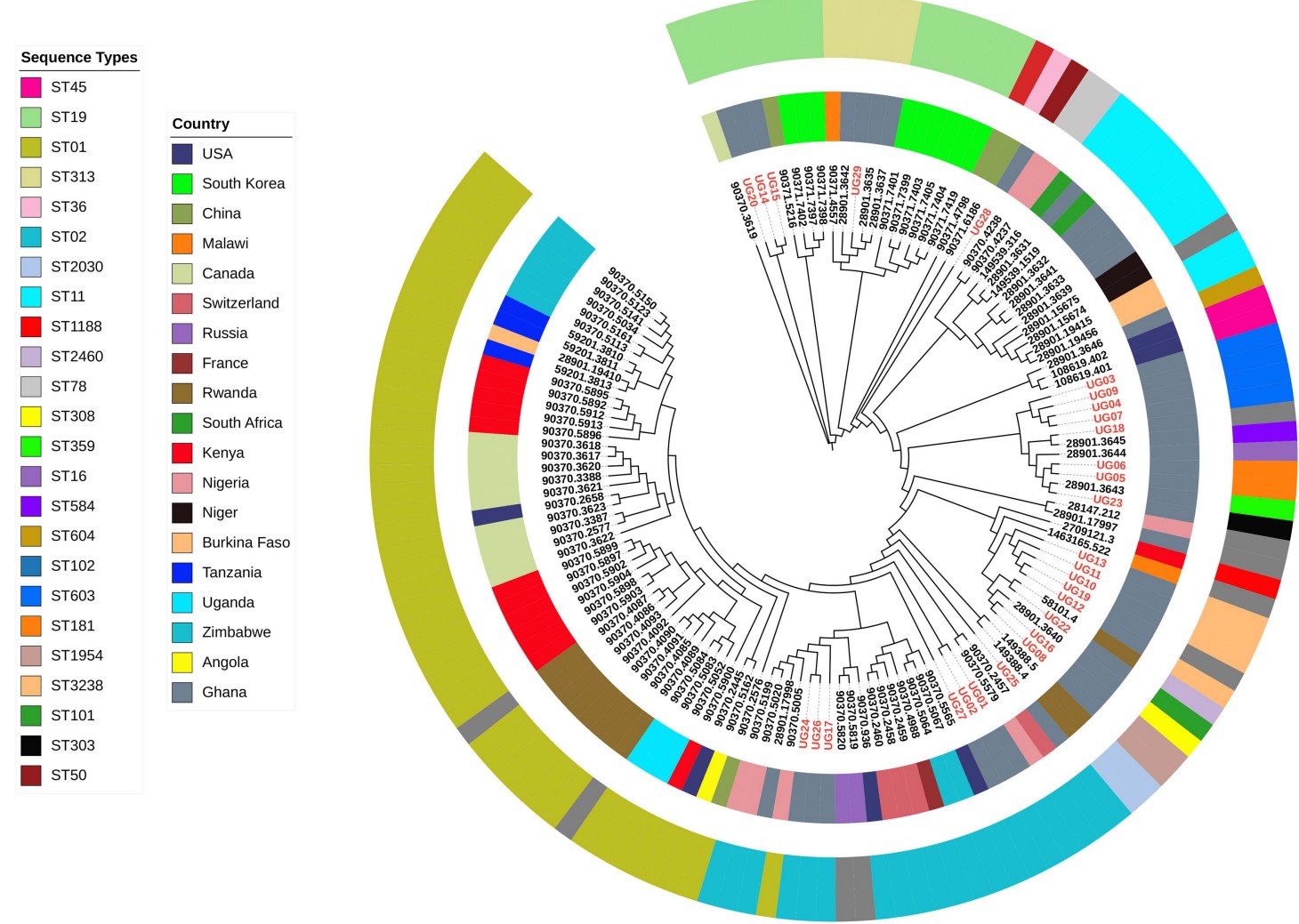

**Fig 1. Core-genome SNP-based phylogenetic analysis of 134 (test strains *n* = 28, local strains *n* = 15, and global strains *n* = 91) publicly available *Salmonella enterica* strains showing diversity among local and global strains.** Inner color key indicating geographical location and outer color key indication STs.

## Virulence factor analysis

*S. enterica* virulence genes were analyzed across the identified serovars (Fig 3, S1 Table). Genes critical for the ability of *Salmonella* to invade host cells including *invA and invE,* were universally present in the serovars. Typhoidal toxin associated genes including *cdtB*, *pltA* and *pltB* were identified in both Typhi serovars and non-typhoidal serovars such as *S.* Durham, *S.* Wein, *S.* Bangui and *S.* Chester. The full length *cdtB* and *pltB* genes were detected in 15/15 (100%) isolates, whiles *pltA* was detected in 14/15 (93.3%) isolates with >97% identity and a high sequence conservation. All the genes were located on the same contig indicating that they reside within a contiguous genomic island. Only one isolate (UG25) displayed fragmented assembly, with relatively shorter contig sizes (~3.1 kb -~3.5 kb), and both *cdtB* and *pltB* appearing on separate contigs, probably explaining the absence of the *pltA* in the blast results (S4 Table in S1 File). T3SS effector genes such as *sopB* and *sopD* showed variability with some serovars lacking one or both. Genes encoding

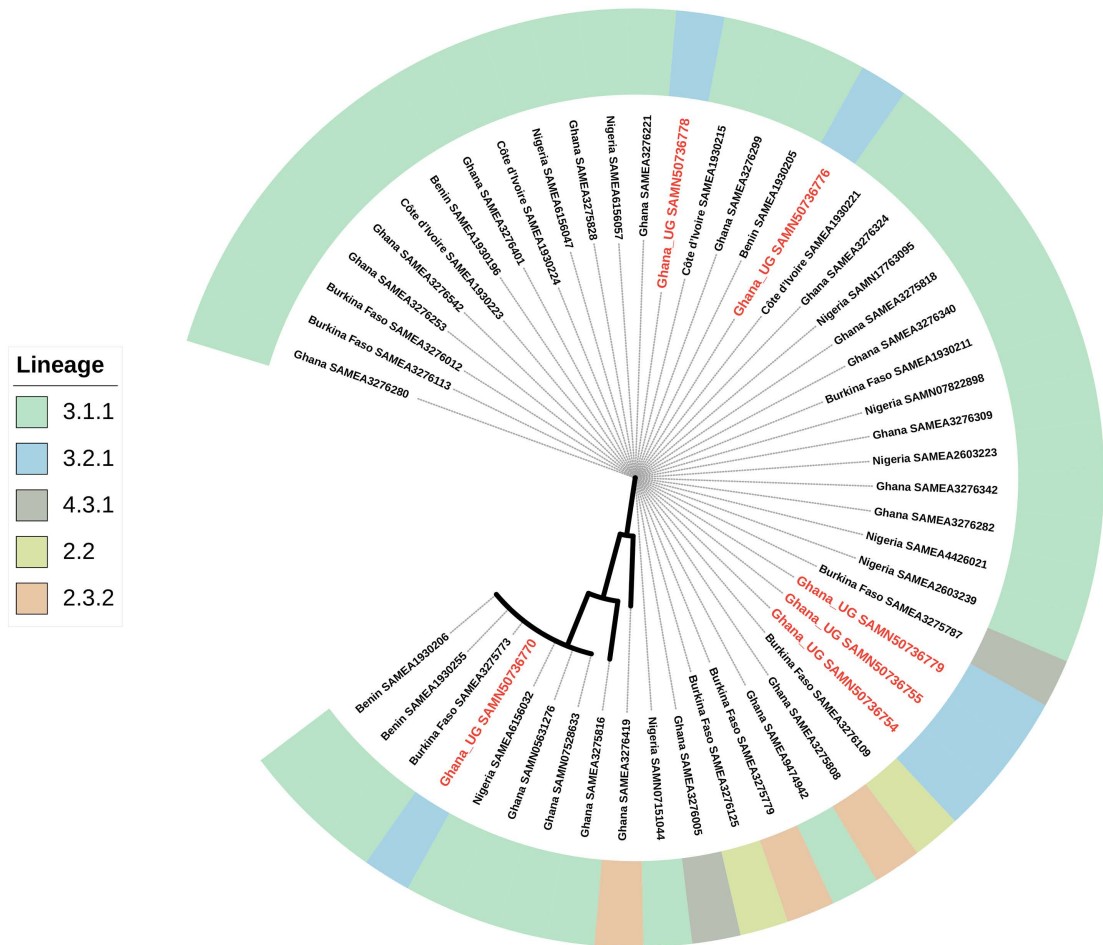

**Fig 2. Unrooted cladogram showing the relationships among *S*. Typhi lineages from Ghana and other West African countries.** Tips are labelled with isolate accession numbers. The outer color strip denotes Genotyphi lineages from this study (Ghana UG sequences) and publicly available genomes from Pathogenwatch used for comparison. 3.1.1 (pale green), 3.2.1 (light blue), 2.3.2 (peach), 2.2 (yellow green), and 4.3.1 (lavender).

adhesion-related proteins, including *fimC* and *csgA*, exhibited serovar-dependent presence, while biofilm formation and intestinal colonization genes (*csgB, ratB,* and *lpfB*) were variably distributed. Intracellular replication genes, such as *mgtC, spvC* and *ssa* operons were largely conserved across isolates, whereas Vi capsule genes (*tviC, vexA, vexD* and *vexE*) were detected in only a subset of serovars including *S*. Typhi and *S*. Bangui.

## Plasmid type analysis and genes associated with antibiotic resistance

PlasmidFinder identified a limited set of plasmid replicon types with marked variation across the different *Salmonella* serovars (Fig 4). Among the *S*. Typhi isolates, only *IncFIB(S)* and *IncFII(S)* plasmids were detected, each occurring in approximately 17% of the isolates. In contrast, *S*. Typhimurium isolates exhibited a broader and more complex plasmid repertoire. All *S*. Typhimurium isolates (100%, n = 4) carried both *IncFIB(S)* and *IncFII(S)*. Additionally, half of the *S*. Typhimurium isolates (50%) harbored the *IncQ1* plasmid, with no *S*. Typhimurium isolate harboring *Col(pHAD28)*, *IncN*, or *IncFIB(K)*. The group representing other *Salmonella enterica* serovars demonstrated the greatest plasmid heterogeneity, though at

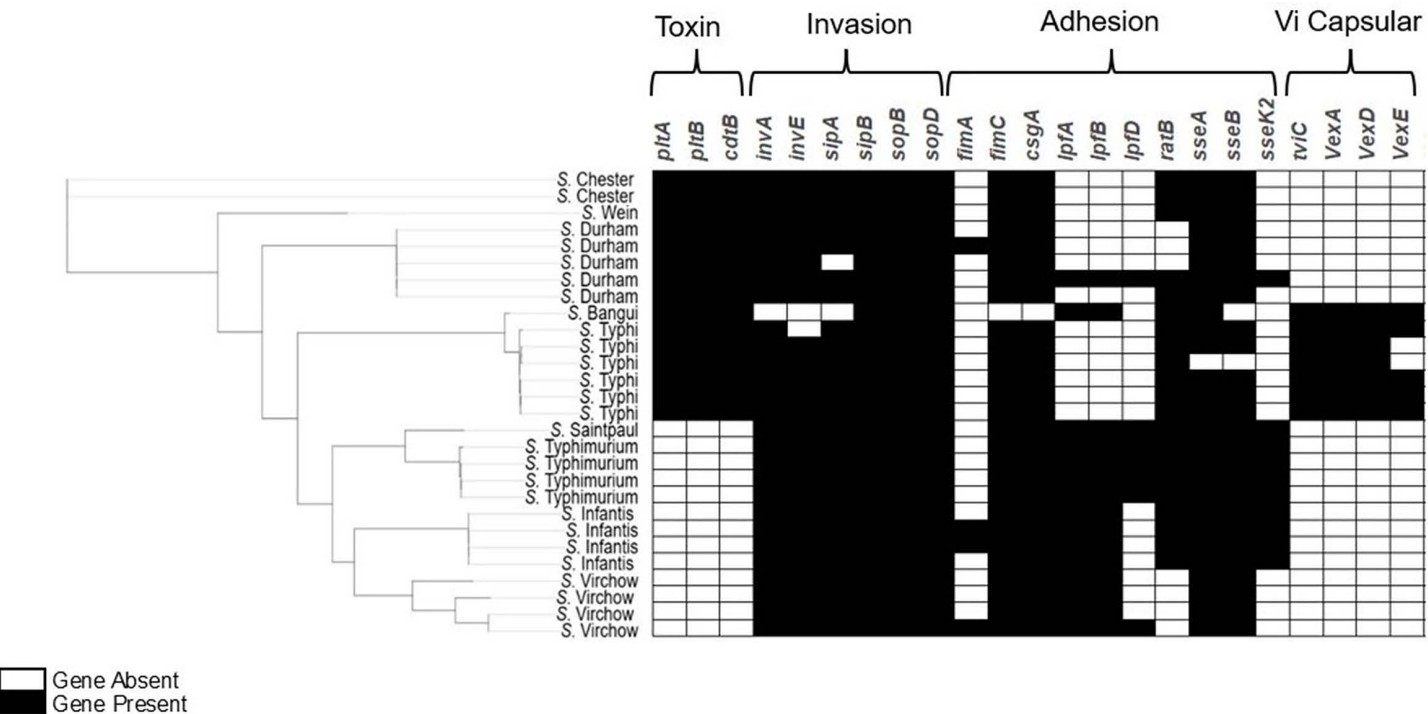

**Fig 3. Heatmap showing the presence (black) and absence (white) of virulence genes across 28 *Salmonella enterica* clinical isolates.** Genes are grouped by functional categories including toxin, invasion, adhesion, and Vi capsular genes. Rows represent individual isolates, and columns represent virulence genes detected using ABRicate against the virulence factor database. The dendrogram on the left indicates hierarchical clustering of isolates.

lower overall frequencies. Replicons identified included *IncFIB(S)* and *IncFII(S)* (each in 10% of isolates), Col(pHAD28) (20%), IncN (10%), IncQ1 (10%), and IncFIB(K) (5%). These plasmids are typically linked with multidrug resistance in *Enterobacteriaceae* [34]. The limited distribution of plasmids across the isolates suggests that, rather than acting as major vehicles for AMR dissemination, plasmids may play a more restricted role in driving resistance evolution in these settings, with other mechanisms or evolutionary events contributing more substantially. The overall detection of antibiotic resistance genes was notably low across isolates with most isolates carrying no resistance genes (Fig 5). Across all *Salmonella* isolates analyzed, the aminoglycoside resistance gene *aac(6')-Iaa* was consistently detected indicating its widespread conservation among the serovars found in the study. Within *S*. Typhi serovars, this gene represented the sole resistance determinant identified, indicating a limited resistance profile restricted to aminoglycoside modification. In contrast, the *S*. Typhimurium isolates (green) displayed a markedly broader resistance gene repertoire. In addition to *aac(6')-Iaa*, they harbored genes associated with resistance to trimethoprim (*dfrA7*) and chloramphenicol (*catA1*), a pattern not observed in all *S*. Typhi and other *S. enterica* serovars. The remaining *S. enterica* serovars (blue) displayed the greatest diversity in antimicrobial resistance determinants. Several isolates in this group possessed multiple quinolone resistance genes (*qnrB10, qnrB36, qnrB5, qnrB61, qnrB82, qnrD*, *qnrS1*, *qnrB19*) indicating a broad capacity for quinolone resistance not seen in *S*. Typhi or *S*. Typhimurium. Furthermore, these serovars were the only group to harbor the tetracycline resistance gene *tetA*. Together, these identified resistance determinants highlight the interplay of antibiotic inactivation, target protection, and potentially reduced permeability or efflux activity. The low prevalence (<20%) of known resistance genes suggests that most of the isolates were probably broadly sensitive to front-line antibiotics, although some of these isolates

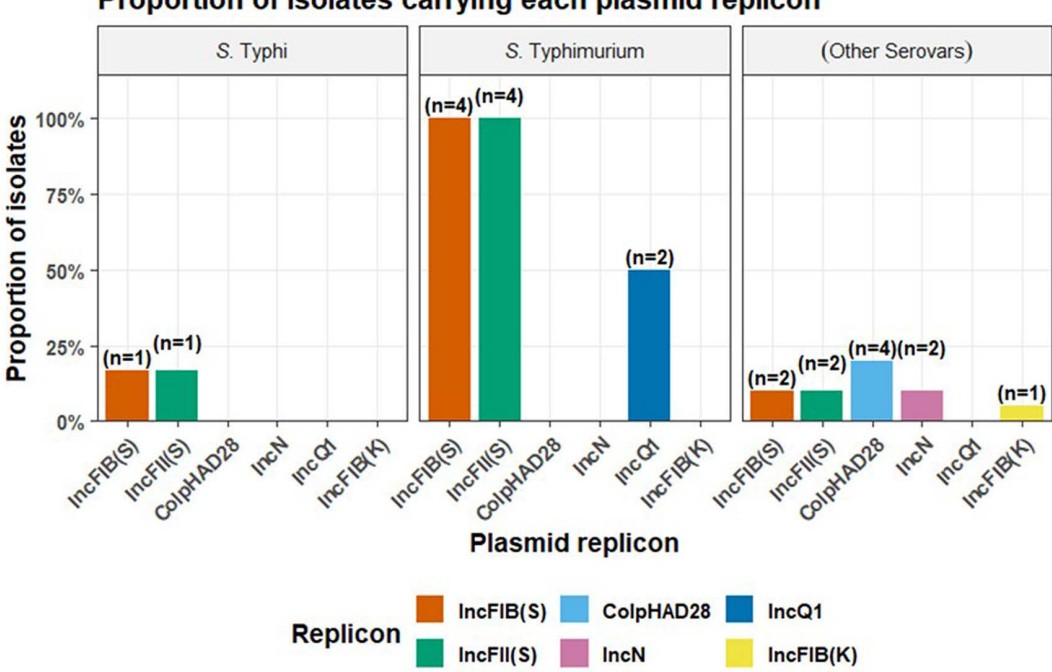

**Fig 4. Faceted bar plots show the proportion of isolates within each serovar group (*S.*Typhi, *S.* Typhimurium, and Other) that carry each plasmid replicon detected by PlasmidFinder.** The replicons (IncFIB(S), IncFII(S), IncN, ColpHAD28, IncFIB(K), IncQ1) are shown by different colored bars, and percentages above bars indicate within-group prevalence.

carried resistance genes in addition to other virulence factors that could potentially confer increased fitness and lead to complications in patient management.

## Discussion

The present study examined the genetic diversity of *S. enterica* strains isolated from patients with a diagnosis of typhoid fever in selected clinical facilities in Ghana. Our analysis focused on strain variability, virulence and genetic markers of antibiotic resistance, with the goal of highlighting *S. enterica* epidemiology, infection outcomes and disease management. Amongst the strains observed, *S.* Typhi ST02 were predominant, which is consistent with most globally reported STs associated with typhoid fever though other regions have reported several other STs within the Typhi serovars [35]. Phylogenetic lineage analysis further demonstrated that all *S.* Typhi isolates belonged to clade 3.2.1. This lineage is distinct from the globally dominant H58 (4.3.1) haplotype and has been reported only sporadically across Africa [36]. Previous studies from West Africa and other global regions have reported the prevalence of H58-associated sub-lineages (3.1.1 and 2.3.2), with no reports of 3.2.1 [35,37,38]. This is consistent with our current dataset, in which all six Ghanaian isolates clustered exclusively within lineage 3.2.1, whereas isolates previously detected in Ghana and other neighboring West African countries grouped under H58-associated lineages 3.1.1,2.2, 4.3.1, and 2.3.2 (Fig 2).

Although, our UG sequences separated across short phylogenetic distances within the tree (Fig 2), classification of lineages is known to be based on canonical single nucleotide polymorphisms (SNPs), which are single diagnostic mutations at defined genomic positions that precisely delineate each lineage [39,40]. In contrast, the phylogenetic tree constructed captures the overall sequence diversity from hundreds of genome-wide SNPs, rather than only the lineage-defining SNPs

**Fig 5. Heatmap illustrates the presence (dark red) and absence (white) of antibiotic resistance genes across twenty-eight (28)** *Salmonella enterica* **isolates.** The rows represent different antibiotic resistance genes grouped by their respective resistance mechanisms (e.g., β-lactamases, quinolone resistance, aminoglycoside-modifying enzymes), while the columns represent individual isolates. Isolates are color-coded by their serovar: red for *S.* Typhi, green for *S.* Typhimurium, and blue for other serovars.

[41]. As such, isolates may appear closely related based on the tree due to shared background SNPs but still belong to distinct lineages defined by their unique canonical SNP profiles.

The detection of 3.2.1 in this study therefore broadens the distribution of circulating *S.* Typhi genotypes in the region, suggesting that both H58 and non-H58 lineages are contributing to endemic transmission. Importantly, unlike H58, lineage 3.2.1 has generally not been associated with extensive multidrug resistance [42,43], which aligns with our previous finding that these isolates remained largely susceptible to first-line antimicrobials [15], and the observed low prevalence of plasmid driven MDR (Fig 5). Thus, while 3.2.1 is a well-defined and globally recognized lineage, its detection in our Ghanaian isolates represents the first report of this genotype.

Two clinically significant Typhimurium serovars, ST19 and ST313 were identified. *S.* Typhimurium ST19, is a global lineage associated with foodborne gastroenteritis in humans [44]. ST313 is a known multidrug resistant clone exhibiting host restricted Typhi-like phenotype [45]. This sequence type has been identified as a predominant cause of NTS bacteremia in most African regions, particularly among febrile and malnourished children [46]. The detection of ST313 in stool samples in this study further aligns with evidence of intestinal carriage as reported in other endemic settings [47]. Both *S.* Typhimurium serovars ST19 and ST313 are known to possess several plasmid-encoded resistance genes and other markers of resistance located on chromosomes [45,48]. Nevertheless, very limited number of core resistance genes and plasmids were identified in these sequence types (Figs 4 and 5). Among the resistance genes detected, *bla*TEM-1B and *bla*TEM-116, which confer resistance to β-lactam antibiotics, were present at relatively low frequencies (14.2%). Additionally, the study identified a handful of quinolone resistance genes (*qnr*B5, *qnr*B19, *qnr*S1) and aminoglycoside-modifying enzymes (*aac* (6')-*Iaa, aadA1*). The fact that these genes were not widespread across the sample set, suggests the

                                                                                 

strains had not yet acquired a significant repertoire of antimicrobial resistance (AMR) determinants. This finding contrasts with the high levels of resistance observed in other regions, where several NTS serovars have acquired multiple resistance genes, leading to multi-drug-resistant strains [49].

The *S. enterica* strains seem to possess a high virulence potential, as indicated by the presence of key virulence factors (Fig 3). Genes such as *invA*, *invE* were consistently present across the isolates, underscoring the potential ability of these strains to effectively breach the intestinal epithelium and initiate more invasive infection [50]. The presence of *sopB* and *sopD* which are involved in the production of T3SS effector proteins that modulate host immune responses, further contributes to the virulence profile by dampening the host's inflammatory response and promoting bacterial survival [51,52]. Sequence types such as ST1954 (*S.* Chester), ST3238 (*S.* Durham, *S.* Jukestown), ST101 (*S.* Wein), and novel STs were obtained from serovars not typically implicated in typhoidal salmonellosis, yet they were found to carry the typhoid toxin genes *cdtB*, *pltA*, and *pltB* (S1 Table and S4 Table in S1 File). These genes are hallmark virulence factors in *S.* Typhi, where they play a key role in immune evasion and systemic infection [53,54]. Except for one isolate (UG25), all three genes were found to be intact. Their presence in these non-typhoidal serovars suggests that these strains may have acquired virulent traits that could potentially enhance their ability to establish persistent infection.

The observed high frequency of other acquired virulence factors in *S. enterica* alongside low detection of AMR, suggests a complex interplay between these two bacterial characteristics. While some studies show a positive correlation between virulence and AMR [55,56], other reports suggests otherwise that the acquisition of virulence factors does not necessarily lead to AMR [57–59]. This dynamic can result in a situation where some *Salmonella* strains are highly efficient in causing infection but less resistant to antibiotics. From a clinical perspective, these findings have significant implications for the management of typhoid fever in these settings in Ghana. Knowing that potentially high virulent non-typhoidal strains may be causing typhoidal symptoms necessitates the need to improve on diagnosis and treatment to mitigate severe disease outcomes. However, the low resistance profile, may provide a window of opportunity to effectively treat infections with existing antibiotics, emphasizing the importance of maintaining the efficacy of these drugs through judicious antibiotic stewardship. Public health strategies should focus on strengthening surveillance systems to detect emerging strains with increased resistance potential while also addressing the underlying factors contributing to the spread of highly virulent *Salmonella* isolates. Given that *Salmonella* bacteria are environmentally transmitted, efforts to improve sanitation, access to clean water, and vaccination coverage are critical to reducing the incidence of typhoid fever and limiting the spread of these virulent strains.

While this study provides valuable insights into the virulence and resistance profiles of *S. enterica* in Ghana, several limitations must be highlighted. The sample size, though informative, may not fully capture the diversity of *Salmonella* strains circulating in the region. Additionally, the study did not explore the functional consequences of the identified virulence genes, and further experimental work will be required to determine whether these virulence loci are expressed and contribute to pathogenic phenotypes or typhoid-like clinical disease. Contrary to these limitations, epidemiological descriptions lean towards a diverse range of *S. enterica* serovars, which are known to cause typhoid fever and other invasive diseases. The diversity of these serovars, coupled with the high virulence potential, underscores the need for continuous surveillance to monitor their spread and evolution within the Ghanaian population. Future studies should also investigate the potential for these strains to acquire resistance determinants through horizontal gene transfer, particularly in the context of increasing antibiotic use in Ghana.

## Conclusions

In summary, this study highlights the genomic diversity of *S. enterica* isolates circulating in Ghana and reveals key features with important clinical and epidemiological implications. While *S.* Typhi isolates were confined to genotype 3.2.1 and carried limited AMR determinants, other non-typhoidal serovars exhibited notable genetic variations and harbored multiple virulence factors, including the typhoid toxin genes (*cdtB, pltA,* and *pltB*). The detection of such toxin

and virulence-associated genes in NTS underscores their potential to contribute to invasive disease beyond classical typhoidal strains. At the same time, the overall absence of extensive multidrug resistance, particularly among the *S.* Typhi 3.2.1 lineage, suggests a different resistance pattern compared to well-documented MDR lineages such as H58. Together, these findings emphasize the need for genomic surveillance to continuously monitor circulating virulent markers including AMR mechanisms to guide antimicrobial stewardship and inform local and regional public health intervention strategies.

## Supporting information

**S1 File. S1, S2, S4 Tables.** Genome assembly QC metrics; estimation of genome completeness and contamination; percent identity, query converge and ORF conservation of the typhoid-like genes.
(PDF)

**S1 Table. Serovars virulome.**
(XLSX)

## Acknowledgments

We extend our sincere gratitude to Mr. Benjamin Ayison, the laboratory scientists, nurses, and staff at St. Luke's Government Hospital, Mercy Women's Catholic Hospital, Saltpond Government Hospital, Asante Akyem Government Hospital, Holy Spirit Clinic, Ho Municipal Hospital, and University Hospital, Legon. Also, the staff of the Next Generation Sequencing Facility at WACCBIP. Their expertise, dedication, and tireless efforts were crucial in the collection and processing of the samples for this study. Their professionalism and commitment to excellence were evident in every aspect of their work, greatly contributing to the success of this research. Their support was not just professional but also inspirational, reminding us of the collaborative spirit essential in advancing scientific knowledge. Thank you for your invaluable contribution to this project.

## Author contributions

**Conceptualization:** Samuel Duodu.

**Data curation:** Derrick A. Daah, Samuel Duodu.

**Formal analysis:** Emmanuel K. Sam, Enock K. Amoako, Collins M. Morang'a.

**Funding acquisition:** Lucas N. Amenga-Etego, Samuel Duodu.

**Investigation:** Emmanuel K. Sam.

**Methodology:** Emmanuel K. Sam, Doreen Akorwome, Righteous K. Agoha.

**Project administration:** Samuel Duodu.

**Resources:** Lucas N. Amenga-Etego, Samuel Duodu.

**Software:** Emmanuel K. Sam, Enock K. Amoako, Collins M. Morang'a, Lucas N. Amenga-Etego.

**Supervision:** Lucas N. Amenga-Etego, Samuel Duodu.

**Validation:** Lucas N. Amenga-Etego, Samuel Duodu.

**Visualization:** Lucas N. Amenga-Etego, Samuel Duodu.

**Writing – original draft:** Emmanuel K. Sam, Derrick A. Daah, Samuel Duodu.

**Writing – review & editing:** Emmanuel K. Sam, Lucas N. Amenga-Etego, Samuel Duodu.

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
