## [Decision Letter · Decision Letter 0]

1 Feb 2026

PONE-D-26-01626Genomic characterization of Salmonella enterica isolates causing typhoid among Ghanaian patientsPLOS One

Dear Dr. Duodu,

Thank you for submitting your manuscript to PLOS ONE. After careful consideration, we feel that it has merit but does not fully meet PLOS ONE’s publication criteria as it currently stands. Therefore, we invite you to submit a revised version of the manuscript that addresses the points raised during the review process.

We look forward to receiving your revised manuscript.

Kind regards,

Gabriel Trueba, PhD

Academic Editor

PLOS One

**Journal Requirements:**

“This research was supported by WACCBIP, University of Ghana, which receives funding from World Bank African Centre of Excellence Masters/PhD fellowship (ACE02-WACCBIP + NCDs, Awandare) and NIHR funded Global Health Research Group on Establishing Regional Hubs for Genomic Surveillance in West Africa grant reference number NIHR13471.”

3. Please note that your Data Availability Statement is currently missing the repository name or the a direct link to access each database. If your manuscript is accepted for publication, you will be asked to provide these details on a very short timeline. We therefore suggest that you provide this information now, though we will not hold up the peer review process if you are unable.

5. Please upload a new copy of Figures 3A and 3B as the detail is not clear. Please follow the link for more information:  https://journals.plos.org/plosone/s/figures

Reviewers' comments:

Reviewer's Responses to Questions

**Comments to the Author**

1. Is the manuscript technically sound, and do the data support the conclusions?

Reviewer #1: Yes

Reviewer #2: Yes

2. Has the statistical analysis been performed appropriately and rigorously? 

Reviewer #1: Yes

Reviewer #2: No

3. Have the authors made all data underlying the findings in their manuscript fully available?

Reviewer #1: Yes

Reviewer #2: Yes

4. Is the manuscript presented in an intelligible fashion and written in standard English?

Reviewer #1: Yes

Reviewer #2: Yes

5. Review Comments to the Author

Reviewer #1: This is a fairy well written manuscript that provides insightful information on S. enterica isolates causing Typhoid fever. It is quite interesting that typhoid toxins genes were found in Non typhoidal isolates. Please find my comments below to improve on the manuscript.

Abstract

The authors should consider adding a statement on the existing research gap this study aimed to fill just before the statement on the research objective in lines 17-19.

The authors should consider qualifying the statement 'Plasmid analysis showed limited diversity. Antibiotic resistance genes were detected at low frequencies, including blaTEM variants (beta-lactam resistance), qnr (quinolone resistance), sul1/sul2 (sulfonamide resistance), aminoglycoside-modifying enzymes (aadA1, aph(6)-Id, aac(3)-IIa), tet(A) (tetracycline resistance), dfrA1 (trimethoprim resistance), and catA1 (chloramphenicol resistance), highlighting limited antimicrobial resistance in the populations sampled' in lines 35-40 by including numbers and proportions of the same.

From the Results section of the abstract, it's not clear, what was the distribution of the S. enterica serotypes after sequencing.

Materials and Methods

The statement 'S. enterica isolates were obtained from patients presenting with typhoid fever in selected health

facilities in Ghana' is not accurate and should be revised to read that the patients presented with typhoid fever symptoms and not typhoid fever unless a typhoid fever positive diagnosis was made at the time of presentation.

In line 101, it's unclear if the S. enterica isolates confirmation was through culture on salmonella shigella agar and if this is the case, this may be inaccurate as colonies on salmonella-shigella are not sufficient enough to give a positive identification of Salmonella spp without further identification using Serology/biochemical testing (API 20E) or molecular techniques such as PCR and sequencing. It would be ideal to refer to the isolates as suspect colonies for Salmonella and not confirmed ones.

Reviewer #2: This manuscript presents WGS-based characterization of 28 Salmonella enterica isolates recovered from stool and blood cultures from patients described as clinically diagnosed with typhoid fever in Ghana. The topic is relevant and the genomic methods are broadly appropriate, but several issues require major revision before the conclusions are reliable.

Major issues

Case definition and sampling frame are unclear

The manuscript frames the work as typhoid fever, yet the dataset includes multiple non-typhoidal Salmonella serovars (e.g., Typhimurium, Enteritidis, Virchow, Saintpaul, etc.). Please clarify:

The clinical definition used to enrol “suspected/diagnosed typhoid” cases (symptoms, duration, exclusion criteria, and whether any diagnostic test beyond culture was used).

The clinical syndrome associated with stool isolates (enteric fever workup vs diarrhoeal illness workup) and whether stool isolates were from the same patients as blood isolates.

A clear breakdown of isolate counts by specimen type (blood vs stool), serovar, and sequence type, ideally in a single table.

If the cohort is “patients suspected of typhoid/enteric fever” rather than culture-confirmed typhoid, this should be reflected consistently in the title, abstract, and conclusions.

Internal inconsistencies in S. Typhi counts and ST reporting

The manuscript reports S. Typhi ST2 as n=7 (24%). However, the supplementary isolate list (Table S1) shows six S. Typhi isolates, all ST2. The seventh ST2 isolate appears to be a non-Typhi serovar (Bangui). This needs to be corrected throughout the text, tables, and figures. Please ensure that all summaries (percentages and n values) match Table S1 and that GenoTyphi lineage assignment is applied to the correct number of S. Typhi genomes.

Virulence gene interpretation is overextended without validation

The report of typhoid toxin genes (cdtB, pltA, pltB) and Vi-related genes outside S. Typhi is potentially interesting, but gene presence calls require more detail and validation because partial hits and fragmented assemblies can produce misleading patterns. In Table S1, several isolates show incomplete toxin gene patterns (e.g., cdtB present without pltA/pltB or vice versa). Please provide:

Identity and coverage thresholds used for calling a gene “present”.

Whether the genes are full-length and intact (no frameshifts or truncations).

Locus context (contig location and surrounding genes, or read-mapping confirmation across the locus).

Without this, discussion implying these non-typhoidal serovars may produce typhoid-like clinical disease should be toned down.

AMR conclusions need phenotypic confirmation or more cautious framing

The manuscript draws strong inferences about susceptibility from genomic AMR gene detection alone. This is not sufficient, especially for fluoroquinolones and other agents where chromosomal mutations and regulatory mechanisms are important. If phenotypic AST was performed, please include it and compare genotype versus phenotype. If AST was not performed, please revise the AMR section to clearly state that findings represent genomic predictions only and avoid clinical treatment implications.

Reproducibility details need strengthening

Please provide software versions and key parameters for the major tools used (read QC and trimming, assembly, annotation, Panaroo settings, IQ-TREE model selection, and thresholds for AMR/virulence detection). Also include basic assembly QC metrics (coverage estimates, N50, contamination checks) and a brief statement on how low-quality assemblies were handled.

Public genome selection for comparative analyses requires justification

The manuscript combines study isolates with public genomes from BV-BRC and Pathogenwatch. Please state explicit inclusion criteria (geography, date range, clinical source, quality filters) and how duplicates or biased sampling were avoided. This affects interpretability of the phylogeographic conclusions.

Minor issues and editorial suggestions

Ensure serovar naming is consistent across the manuscript and supplementary material (avoid spacing/format variants that can affect counts).

Improve figure legends so a reader can interpret rings/labels without referring back to methods.

Consider adding a simple flow diagram of recruitment and isolate selection (patients enrolled, cultures done, positives, exclusions, final 28 isolates).

6. PLOS authors have the option to publish the peer review history of their article (what does this mean?). If published, this will include your full peer review and any attached files.

Reviewer #1: No

Reviewer #2: **Yes:** Nubwa Medugu

---

## [Author Response · Author response to Decision Letter 1]

9 Mar 2026

Editorial comments

Response:

The manuscript has been realigned to meet PLOS ONE style requirements.

“This research was supported by WACCBIP, University of Ghana, which receives funding from World Bank African Centre of Excellence Masters/PhD fellowship (ACE02-WACCBIP + NCDs, Awandare) and NIHR funded Global Health Research Group on Establishing Regional Hubs for Genomic Surveillance in West Africa grant reference number NIHR13471.”

Response:

The financial disclosure statement has been revised to “This research was supported by WACCBIP, University of Ghana, which receives funding from World Bank African Centre of Excellence Masters/PhD fellowship (ACE02-WACCBIP + NCDs, Awandare) and NIHR funded Global Health Research Group on Establishing Regional Hubs for Genomic Surveillance in West Africa grant reference number NIHR13471. The funders had no role in study design, data collection and analysis, decision to publish, or preparation of the manuscript."

3. Please note that your Data Availability Statement is currently missing the repository name or the a direct link to access each database. If your manuscript is accepted for publication, you will be asked to provide these details on a very short timeline. We therefore suggest that you provide this information now, though we will not hold up the peer review process if you are unable.

Response:

The genome sequences have been deposited in NCBI and the direct link to access the data can be found at https://www.ncbi.nlm.nih.gov/bioproject/PRJNA1309152

Response:

Ethics statement details have been included in the “Methods” section.

5. Please upload a new copy of Figures 3A and 3B as the detail is not clear. Please follow the link for more information: https://journals.plos.org/plosone/s/figures

Response:

We have uploaded new figures for 3A and 3B.

Response:

No recommendation was made by reviewers with regards to specific citation of previous works.

Reviewer #1

This is a fairy well written manuscript that provides insightful information on S. enterica isolates causing Typhoid fever. It is quite interesting that typhoid toxins genes were found in Non typhoidal isolates. Please find my comments below to improve on the manuscript.

Abstract

The authors should consider adding a statement on the existing research gap this study aimed to fill just before the statement on the research objective in lines 17-19.

Response:

A brief statement highlighting the existing research gap addressed by this study has been added immediately before the research objective in lines 17–19.

The authors should consider qualifying the statement 'Plasmid analysis showed limited diversity. Antibiotic resistance genes were detected at low frequencies, including blaTEM variants (beta-lactam resistance), qnr (quinolone resistance), sul1/sul2 (sulfonamide resistance), aminoglycoside-modifying enzymes (aadA1, aph(6)-Id, aac(3)-IIa), tet(A) (tetracycline resistance), dfrA1 (trimethoprim resistance), and catA1 (chloramphenicol resistance), highlighting limited antimicrobial resistance in the populations sampled' in lines 35-40 by including numbers and proportions of the same.

Response:

The statement has been revised to include the number and proportion of isolates carrying each plasmid type and antimicrobial resistance gene, providing clearer quantitative context to support the conclusion of limited antimicrobial resistance in the populations sampled.

From the Results section of the abstract, it's not clear, what was the distribution of the S. enterica serotypes after sequencing.

Response:

The abstract has been updated to clearly state the distribution of S. enterica serotypes identified following whole-genome sequencing.

Materials and Methods

The statement 'S. enterica isolates were obtained from patients presenting with typhoid fever in selected health facilities in Ghana' is not accurate and should be revised to read that the patients presented with typhoid fever symptoms and not typhoid fever unless a typhoid fever positive diagnosis was made at the time of presentation.

Response:

We clarify that patients were diagnosed at the participating hospitals based on clinical presentation consistent with typhoid fever together with routine serological testing, in line with standard practice at the study sites. As such, the phrase “presenting with typhoid fever” reflects a clinical plus serology-based diagnosis at presentation, rather than symptoms alone. This diagnostic framework has been described in detail in our previously published study from the same patient population (Sam et al, BMC Infect Dis. 2024;24(1):1262).

In line 101, it's unclear if the S. enterica isolates confirmation was through culture on salmonella shigella agar and if this is the case, this may be inaccurate as colonies on salmonella-shigella are not sufficient enough to give a positive identification of Salmonella spp without further identification using Serology/biochemical testing (API 20E) or molecular techniques such as PCR and sequencing. It would be ideal to refer to the isolates as suspect colonies for Salmonella and not confirmed ones.

Response:

We confirm that Salmonella-Shigella agar was used only for the initial isolation of suspect colonies, and not for definitive identification. All isolates were subsequently confirmed using standard biochemical testing, including API 20E, and further validated through molecular characterization as described in our previous study from which these isolates were derived. We agree that selective media alone is insufficient for confirmation, and the identification workflow followed a stepwise approach consistent with standard microbiological practice.

Reviewer #2

This manuscript presents WGS-based characterization of 28 Salmonella enterica isolates recovered from stool and blood cultures from patients described as clinically diagnosed with typhoid fever in Ghana. The topic is relevant and the genomic methods are broadly appropriate, but several issues require major revision before the conclusions are reliable.

Major issues

Case definition and sampling frame are unclear

The manuscript frames the work as typhoid fever, yet the dataset includes multiple non-typhoidal Salmonella serovars (e.g., Typhimurium, Enteritidis, Virchow, Saintpaul, etc.). Please clarify:

Response:

Patients were enrolled based on clinical suspicion of typhoid fever (case definition), which reflects routine practice in the participating health facilities. Although S. Typhi is the classical cause of typhoid, other serovars (S. Paratyphi and S. Typhimurium) are known to present typhoidal symptoms. The detection of other serovars may reflect the overlap in clinical presentation of typhoidal and non-typhoidal Salmonella infections in this setting, rather than an inconsistency in the sampling framework.

The clinical definition used to enrol “suspected/diagnosed typhoid” cases (symptoms, duration, exclusion criteria, and whether any diagnostic test beyond culture was used).

Response:

The clinical definition used for enrolment of suspected cases of typhoid was well described in our previously published study, from which isolates used in this study were obtained. For all patients recruited into the study, culture and serological testing were performed in line with standard practice for typhoid diagnosis.

The clinical syndrome associated with stool isolates (enteric fever workup vs diarrhoeal illness workup) and whether stool isolates were from the same patients as blood isolates.

A clear breakdown of isolate counts by specimen type (blood vs stool), serovar, and sequence type, ideally in a single table

Response:

We thank the reviewer for this comment. The clinical presentation, diagnostic workup (enteric fever versus diarrhoeal illness), and patient-level linkage of blood and stool isolates were comprehensively described in our previously published study (Sam et al., 2024), from which the isolates analyzed here were derived. Specifically, that paper reports culture recovery by specimen type (blood vs stool) and the isolation methods set that were used for downstream analyses. The present manuscript focuses specifically on downstream whole-genome sequencing and molecular characterization of recovered Salmonella enterica isolates; therefore, detailed clinical stratification was not repeated. However, the breakdown of isolate counts by specimen type (blood vs stool), serovar and sequence type is provided in the supporting information S3 Table.

If the cohort is “patients suspected of typhoid/enteric fever” rather than culture-confirmed typhoid, reflect consistently in title, abstract, conclusions.

Response:

We respectfully clarify that patients were managed at presentation as typhoid/enteric fever cases based on routine clinical assessment and serological testing, and isolates included in the present study were cultured and confirmed as Salmonella enterica, originating from the same study cohort as reported in Sam et al., 2024.

Internal inconsistencies in S. Typhi counts and ST reporting.

The manuscript reports S. Typhi ST2 as n=7 (24%). However, the supplementary isolate list (Table S1) shows six S. Typhi isolates, all ST2. The seventh ST2 isolate appears to be a non-Typhi serovar (Bangui). This needs to be corrected throughout the text, tables, and figures. Please ensure that all summaries (percentages and n values) match Table S1 and that GenoTyphi lineage assignment is applied to the correct number of S. Typhi genomes.

Response:

We thank the reviewer for identifying this inconsistency. Upon re-examination of Table S1 (now S3 Table), we confirm that six isolates are Salmonella Typhi, all belonging to ST2, while the additional ST2 isolate corresponds to a non-Typhi serovar (S. Bangui). We have corrected this throughout the manuscript, tables, figures, and percentage calculations.

Virulence gene interpretation is overextended without validation

The report of typhoid toxin genes (cdtB, pltA, pltB) and Vi-related genes outside S. Typhi is potentially interesting, but gene presence calls require more detail and validation because partial hits and fragmented assemblies can produce misleading patterns. In Table S1, several isolates show incomplete toxin gene patterns (e.g., cdtB present without pltA/pltB or vice versa). Please provide: Identity and coverage thresholds used for calling a gene “present”. Whether the genes are full-length and intact (no frameshifts or truncations). Locus context (contig location and surrounding genes, or read-mapping confirmation across the locus). Without this, discussion implying these non-typhoidal serovars may produce typhoid-like clinical disease should be toned down.

Response:

We agree that virulence gene detection based on short-read assemblies can be influenced by assembly fragmentation. We have therefore provided a detailed analysis of the data to address this concern. We found that except in one case, the typhoid toxin-associated genes (cdtB, pltA, and pltB) were consistently detected as a “complete gene set” when present. This has been corrected in the supplementary Table S1 (now S3 Table) and throughout the manuscript. Detailed analysis of these genes is also done and information on their contig location, percent identity, query converge and intactness is provided (S4 Table).

AMR conclusions need phenotypic confirmation or more cautious framing

The manuscript draws strong inferences about susceptibility from genomic AMR gene detection alone. This is not sufficient, especially for fluoroquinolones and other agents where chromosomal mutations and regulatory mechanisms are important. If phenotypic AST was performed, please include it and compare genotype versus phenotype. If AST was not performed, please revise the AMR section to clearly state that findings represent genomic predictions only and avoid clinical treatment implications.

Response:

We would like to clarify that phenotypic antimicrobial susceptibility testing has already been performed and reported by our group in a previously published study from the same study cohort (Sam et al., 2024), using CLSI-guided disk diffusion methods, which is duly referenced in this manuscript. The phenotypic resistance patterns observed in our previous paper are consistent with the genomic AMR profiles described in the present manuscript, including the generally low levels of resistance detected. Accordingly, our interpretation of AMR findings in the current study is strengthened by this prior genotype-phenotype concordance, although we recognize that the present manuscript itself focuses on genomic analyses.

Reproducibility details need strengthening

Please provide software versions and key parameters for the major tools used (read QC and trimming, assembly, annotation, Panaroo settings, IQ-TREE model selection, and thresholds for AMR/virulence detection). Also include basic assembly QC metrics (coverage estimates, N50, contamination checks) and a brief statement on how low-quality assemblies were handled.

Response:

Reproducibility details have been strengthened by explicitly reporting software versions and key parameters for all major tools used in reads QC, trimming, assembly, annotation, and downstream analyses. Assembly quality assessment and handling of low-quality assemblies are now clearly stated in the Methods. Isolates were screened for genome completeness and contamination using CheckM (v1.2.4). Assemblies were initially required to meet the high-quality standard (> 90% completeness and < 5 % contamination). All the samples met this quality threshold except one sample UG25, which initially exhibited high strain heterogeneity (89.3%) and contamination (44.6 %), possibly due to a co-assembly of closely related Enterobacteriaceae. To ensure data integrity, a length-based filtering protocol was implemented and only contigs ≥1,500bp were retained. This refined the assembly to a medium quality draft with 74.6% completeness and 4.55% contamination, meeting the required safety threshold for subsequent comparative genomic analyses.

Public genome selection for comparative analyses requires justification

The manuscript combines study isolates with public genomes from BV-BRC and Pathogenwatch. Please state explicit inclusion criteria (geography, date range, clinical source, quality filters) and how duplicates or biased sampling were avoided. This affects interpretability of phylogeographic conclusions.

Response:

We acknowledge that, as with most publicly available datasets, some degree of regional sampling imbalance is inherent to database availability. However, the inclusion criteria and uniform analytical framework applied here were designed to maximize interpretability and robustness of the inferred phylogeographic relationships. Firstly, public genomes were obtained from the PATRIC–BV-BRC database, restricted to human-derived Salmonella enterica isolates. Only high-quality, complete genomes were included to ensure consistency and minimize assembly-related biases in downstream pangenome and phylogenetic analyses. Genomes lacking essential metadata or showing poor assembly quality were excluded. Moreover,

---

## [Decision Letter · Decision Letter 1]

26 Apr 2026

Genomic characterization of Salmonella enterica isolates causing typhoid among Ghanaian patients

PONE-D-26-01626R1

Dear Dr. Duodu,

We’re pleased to inform you that your manuscript has been judged scientifically suitable for publication and will be formally accepted for publication once it meets all outstanding technical requirements.

Kind regards,

Gabriel Trueba, PhD

Academic Editor

PLOS One

Additional Editor Comments (optional):

Reviewers' comments:

Reviewer's Responses to Questions

**Comments to the Author**

1. If the authors have adequately addressed your comments raised in a previous round of review and you feel that this manuscript is now acceptable for publication, you may indicate that here to bypass the “Comments to the Author” section, enter your conflict of interest statement in the “Confidential to Editor” section, and submit your "Accept" recommendation.

Reviewer #2: All comments have been addressed

Reviewer #3: All comments have been addressed

2. Is the manuscript technically sound, and do the data support the conclusions?

Reviewer #2: Yes

Reviewer #3: Yes

3. Has the statistical analysis been performed appropriately and rigorously? 

Reviewer #2: Yes

Reviewer #3: Yes

4. Have the authors made all data underlying the findings in their manuscript fully available?

Reviewer #2: Yes

Reviewer #3: Yes

5. Is the manuscript presented in an intelligible fashion and written in standard English?

Reviewer #2: Yes

Reviewer #3: Yes

6. Review Comments to the Author

Reviewer #2: The authors have satisfactorily addressed my previous comments. The revised manuscript is substantially improved, and the main concerns raised during the earlier round of review have been resolved. The manuscript is technically sound, the analyses are adequately described, and the presentation is clear. I consider the manuscript acceptable for publication in its current form.

Reviewer #3: (No Response)

7. PLOS authors have the option to publish the peer review history of their article (what does this mean?). If published, this will include your full peer review and any attached files.

Reviewer #2: **Yes:** Nubwa Medugu

Reviewer #3: **Yes:** Samuel Kariuki

---

## [Editor Report · Acceptance letter]

PONE-D-26-01626R1

PLOS One

Dear Dr. Duodu,

I'm pleased to inform you that your manuscript has been deemed suitable for publication in PLOS One. Congratulations! Your manuscript is now being handed over to our production team.

Kind regards,

on behalf of

Dr. Gabriel Trueba

Academic Editor

PLOS One